# Improving End-to-End Training of Retrieval-Augmented Generation Models via Joint Stochastic Approximation

## Abstract

Retrieval-augmented generation (RAG) has become a widely recognized paradigm to combine parametric memory with non-parametric memory. An RAG model consists of two serial connecting components (retriever and generator). A major challenge in end-to-end optimization of the RAG model is that marginalization over relevant passages (modeled as discrete latent variables) from a knowledge base is required. Traditional top-K marginalization and variational RAG (VRAG) suffer from biased or high-variance gradient estimates. In this paper, we propose and develop joint stochastic approximation (JSA) based end-to-end training of RAG, which is referred to as JSA-RAG. The JSA algorithm is a stochastic extension of the EM (expectation-maximization) algorithm and is particularly powerful in estimating discrete latent variable models. Extensive experiments are conducted on five datasets for two tasks (open-domain question answering, knowledge-grounded dialogs) and show that JSA-RAG significantly outperforms both vanilla RAG and VRAG. Further analysis shows the efficacy of JSA-RAG from the perspectives of generation, retrieval, and low-variance gradient estimate.

## 1 Introduction

Large language models (LLMs) have been shown to store factual knowledge in their parameters through pre-training over large amounts of Internet corpora (Petroni et al., 2019; Brown et al., 2020). However, such implicit knowledge cannot be easily updated, expanded, inspected, and interpreted. Moreover, for many knowledge-intensive tasks, the use of external knowledge beyond the parametric memory of LLMs to generate responses is critical, such as in open-domain question answering (ODQA) (Chen et al., 2017; Lee et al., 2019; Karpukhin et al., 2020) and knowledge-grouned dialog systems (Kim et al.; Mishra et al., 2022; Cai et al., 2023). To address these issues, hybrid models that combine parametric memory with nonparametric memories have emerged (Lee et al., 2019; Guu et al., 2020), among which retrieval-augmented generation (RAG) has drawn considerable attention (Lewis et al., 2020).

During recent years, RAG has not only been used to refer to the particular method developed in (Lewis et al., 2020), but also, more often, represents a general two-step paradigm (retrieve-then-generate). In the RAG paradigm, given a context (denoted by $x$) such as a query in QA or a dialog context, relevant passages (denoted by $h$) are first obtained from external knowledge bases (KBs) by using a retriever. The retrieved passages are then combined with the context and fed into a generator to generate the response $y$.

Hence, a RAG model consists of two serial connecting components (retriever and generator). During training, if the relevant passage is known (e.g., human-annotated gold passage), we can supervise the retriever with that passage, and train the generator conditioned on that passage as well. However, collecting human annotations of gold passages is labor intensive. This challenge leads to the widely adopted approach of training retrievers and generators separately. Retrievers are often trained on one corpus (Karpukhin et al., 2020; Izacard et al.; Zhang et al., 2023), and then generators are trained on another different corpus using fixed retrievers (Khattab et al., 2022; Zhang et al., 2024; Zhao et al., 2024). While this is fairly easy to implement, separate training is sub-optimal, for example, the retriever never improves as the generator learns to generate responses.

There have been efforts in developing end-to-end training of an RAG model (Lewis et al., 2020; Zhang et al., 2022; Han et al., 2023; Zamani & Bendersky, 2024), which means eliminating the reliance on intermediate annotations and training all model components simultaneously. In (Lewis et al., 2020), to train the retriever and generator end-to-end, the relevant passage is treated as a discrete latent variable and the following marginal log-likelihood is to be maximized:

$$p(y|x) = \sum_h p(h|x)p(y|x,h) \tag{1}$$

Thus end-to-end training of RAG in essence amounts to unsupervised training of a discrete latent-variable model, as shown above. Direct marginalization is intractable; hence, originally, top-K marginalization (TKM) is used for the approximation (Lewis et al., 2020), which we refer to as vanilla RAG. Recently, variational learning (VL) (Kingma & Welling, 2014) has been applied to end-to-end training of RAG in two concurrent and similar works - VRAG (Mishra et al., 2022) and Hindsight (Paranjape et al.), which we refer to collectively as VRAG. In VRAG, an auxiliary inference model is introduced, acting as a posterior retriever. However, for variational learning of discrete latent variable models, the traditional Monte Carlo gradient estimator for the inference model parameter is known to be either biased or have high-variance (Ou & Song, 2020).

Recently, the joint stochastic approximation (JSA) algorithm (Xu & Ou, 2016; Ou & Song, 2020) has emerged to learn discrete latent variable models with better performance than VL. JSA is a stochastic extension of the EM (expectation-maximization) algorithm and gives unbiased, low-variance stochastic gradients for the inference model.

In this paper, we propose JSA based end-to-end training of RAG, which is referred to as JSA-RAG, as overviewed in Figure 1. JSA-RAG makes the following contributions. First, we design all model components (including prior retriever, generator, and posterior retriever) and implement the whole training and decoding pipeline to enable the successful application of JSA. We address some computational challenges to work with large-scale KBs (e.g., tens of millions of passages in Wikipedia). Second, we investigate the effect of index rebuilding in training. We study the passage concatenation strategy for post-training of the generator while fixing the retriever. These further demonstrate the capability and bonus offered by JSA-RAG. Third, extensive experiments are conducted on five datasets for two tasks (open-domain question answering, knowledge-grounded dialogs) and show that JSA-RAG outperforms both vanilla RAG and VRAG, e.g., achieving +4.1% Exact Match on TQA and +10.3% BLEU-4 on DoQA relative over VRAG. Improved retriever performance and low-variance gradients of the posterior retriever are also validated, e.g.,+8.5% R@1 on NQ and +1.7% R@1 on OR-QuAC relative over VRAG.

## 2 METHOD: JSA-RAG

### 2.1 MODEL

Let $(x, y)$ denote the pair of context and response, both represented by token sequences. Let $\mathcal{K}$ denote the KB, which is a discrete set of text passages (e.g. Wikipedia chunks). Each passage is also a token sequence. Let $h$ denote the relevant passage in $\mathcal{K}$ needed to generate the response $y$ given the context $x$, which is treated as a latent variable since there are no annotations. Therefore, we obtain a latent variable model for RAG, with parameters $\theta = (\theta_r, \theta_g)$, which can be decomposed as:

$$p_\theta(y, h|x) = p_{\theta_r}(h|x)p_{\theta_g}(y|x, h) \tag{2}$$

**Prior retriever** $p_{\theta_r}(h|x)$, is parameterized by $\theta_r$ and models the prior relevancy of the passages in $\mathcal{K}$ with respect to the context $x$. Similar to the original RAG, a bi-encoder architecture for the prior retriever is defined as follows:

$$p_{\theta_r}(h|x) = \frac{\exp\left(\mathbf{e}_\lambda(h)^\top \mathbf{e}_\eta(x)\right)}{\sum_{h' \in \mathcal{K}} \exp\left(\mathbf{e}_\lambda(h')^\top \mathbf{e}_\eta(x)\right)} \tag{3}$$

where $\mathbf{e}_\lambda(x)$ denote the context encoder, parameterized by $\lambda$, outputting the dense vector representation (or say, the embedding vector) of the context; $\mathbf{e}_\eta(h)$, the passage encoder, parameterized by $\eta$, returns the embedding vector of the passage. We calculate the two embeddings with two separate neural networks, both initialized from BERT (Devlin et al., 2019). Hence, $\theta_r = (\lambda, \eta)$.

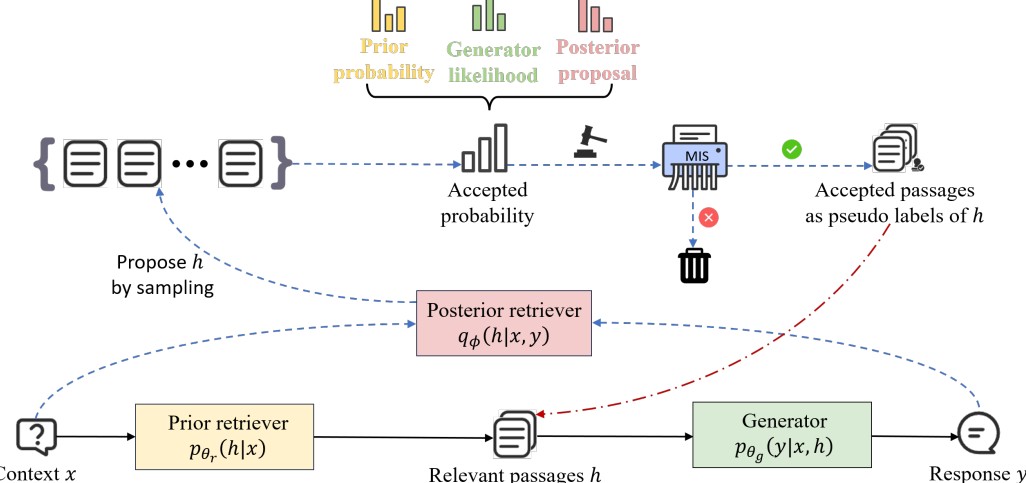

Figure 1: Overview of JSA-RAG. 1) In addition to the (prior) retriever and generator, JSA-RAG introduces an (auxiliary) posterior retriever. 2) During training, the posterior retriever proposes relevant passages, which get accepted or rejected according to the probabilities calculated from the three components. The blue dashed line shows such Metropolis independence sampling (MIS), which is a Monte Carlo approximation of the E-step in EM. 3) The filtered passages are then treated as pseudo labels, as shown by the red dotted line. 4) Given the pseudo labels, we can calculate the gradients for prior retriever, posterior retriever, and generator, respectively, and proceed with parameter updating, very similar to perform supervised training, like the M-step in EM.

**Generator** $p_{\theta_g}(y|x, h)$ is parameterized by $\theta_g$ and models the sequential generation of the response $y$ given the context $x$ and the passage $h$. The neural network architecture can be encoder-decoder or decodely-only. In this work, we employ decoder-only LLMs, which calculates the likelihood of the response $y$ as follows:

$$\log p(y|h, x) = \sum_j \log p(y_j|y_{<j}, x, h). \tag{4}$$

where the context $x$ and the retrieved passage $h$ are concatenated to fed into the LLM to generate $y$.

**Posterior retriever** is introduced for applying the JSA algorithm to learn the latent variable model Eq. (2). It represents an auxiliary inference model to approximate the posterior probability of selecting passage $h$ when given both context $x$ and reponse $y$. Similar to the prior retriever, a bi-encoder architecture for the posterior retriever, with parameters $\phi = (\lambda, \xi)$, is defined as follows:

$$q_\phi(h|x, y) = \frac{\exp\left(\mathbf{e}_\lambda(h)^\top \mathbf{e}_\xi(x + y)\right)}{\sum_{h' \in \mathcal{K}} \exp\left(\mathbf{e}_\lambda(h')^\top \mathbf{e}_\xi(x + y)\right)} \tag{5}$$

where the passage encoder $\mathbf{e}_\lambda(h)$ is shared between the prior and the posterior retrievers, but a new BERT based neural network is introduced to calculate the embedding for the combination of context $x$ and response $y$. In particular, $x$ and $y$ are concatenated, denoted by $x + y$, and are fed to the context-response encoder $\mathbf{e}_\xi(\cdot)$. Note that except for the index rebuilding experiment, all passage encoders are fixed.

**Computation consideration.** The softmax calculation over the entire KB in Eq. (3) and Eq. (5) for the prior and posterior retrievers are computationally prohibitive even for moderate-sized KBs (e.g., thousands of passages). In practice, we maintain an index on passage embeddings for the KB using FAISS (Johnson et al., 2019). Given a pair of context and reponse $(x, y)$, we can efficiently retrieve the set of top-$k$ passages under prior and posterior distributions using Maximum Inner Product Search (MIPS) (Johnson et al., 2019), denoted by $\mathcal{S}^{\text{prior}}$ and $\mathcal{S}^{\text{post}}$ respectively ($k = 10$ in our experiments). The two sets occupy the majority of probabilities for the prior and posterior distributions, and at first thought, can be used to approximate the calculations of Eq. (3) and Eq. (5), respectively. Note that in order to calculate the importance weights for sampled passages used in JSA training (to be clear in Section 2.2 below), we need the prior and posterior probabilities to be calculated over a common set. Therefore, we form a union set by merging $\mathcal{S}^{\text{prior}}$ and $\mathcal{S}^{\text{post}}$, and the softmax calculations in Eq. (3) and Eq. (5) are only taken over this union set.

---

**Algorithm 1** The JSA-RAG algorithm

---

**Require:** Training dataset $\mathcal{D} = \{(x, y)\}$, prior retriever $p_{\theta_r}(h|x)$, posterior retriever $q_\phi(h|x, y)$, generator $p_{\theta_g}(y|x, h)$, MIS step number $m$.
  **repeat**
    Draw a pair of context and response $(x, y)$;
    **Monte Carlo sampling:**
    Use MIS to draw $\{h^{(1)}, h^{(2)}, ..., h^{(m)}\}$;
    **Parameter updating:**
    Update $\theta$ by ascending:
    $\frac{1}{m} \sum_{i=1}^{m} \nabla_\theta \log \left[ p_{\theta_r}(h^{(i)}|x) p_{\theta_g}(y|x, h^{(i)}) \right]$;
    Update $\phi$ by ascending:
    $\frac{1}{m} \sum_{i=1}^{m} \nabla_\phi \log q_\phi(h^{(i)}|x, y)$;
  **until** convergence
  **return** $\theta$ and $\phi$

---

## 2.2 TRAINING

Training the RAG model from complete data, i.e., knowing $h$, can be easily realized by supervised training. For end-to-end training of the RAG model (i.e., conducting unsupervised training without knowing $h$), we resort to maximizing the marginal likelihood $p_\theta(y|x)$ and applying the JSA algorithm (Xu & Ou, 2016; Ou & Song, 2020).

JSA involves introducing an auxiliary inference model to approximate the intractable posterior $p_\theta(h|x, y)$, which, turns out to take the form of $q_\phi(h|x, y)$, i.e., the posterior retriever. We can jointly train the three components (prior retriever, posterior retriever and generator), which is summarized in Algorithm 1. The JSA algorithm can be viewed as a stochastic extension of the well-known EM algorithm (Dempster et al., 1977) , which iterates Markov Chain Monte Carlo (MCMC) sampling and parameter updating, being analogous to the E-step and the M-step in EM respectively.

**E-Step.** The sampling step stochastically retrieves passages through sampling from the posterior $p_\theta(h|x, y)$. However, direct sampling from the posterior $p_\theta(h|x, y)$ is intractable, so MCMC sampling is employed. Particularly, using $p_\theta(h|x, y)$ as the target distribution and $q_\phi(h|x, y)$ as the proposal, we sample $h$ through Metropolis independence sampler (MIS) (Liu, 2001) as follows:

1) Propose $h \sim q_\phi(h|x, y)$;

2) Accept $h$ with probability $\min \left\{ 1, \frac{w(h)}{w(\tilde{h})} \right\}$, where

$$w(h) = \frac{p_\theta(h|x, y)}{q_\phi(h|x, y)} \propto \frac{p_{\theta_r}(h|x) p_{\theta_g}(y|x, h)}{q_\phi(h|x, y)} \tag{6}$$

is the usual importance ratio between the target and the proposal distribution and $\tilde{h}$ denotes the previous value for $h$ along the Markov chain. In practice, we run MIS for several $(m)$ steps, with the chain is initialized from $p_\theta(h|x, y)$.

**M-Step.** Once we obtain the accepted pseudo labels $\{h^{(1)}, h^{(2)}, ..., h^{(m)}\}$ from MIS, we can treat them as being given. We calculate the gradients for the prior retriever, posterior retriever, and generator models, respectively, and proceed with parameter updating, very similar to the process in supervised training. This is analogous to the M-step in EM, but the proposal $q_\phi$ is also adapted. In summary, the loss function can be written as:

$$\mathcal{L}_{\text{JSA}} = -\frac{1}{m} \sum_{i=1}^{m} \left( \log p_{\theta_r}(h^{(i)}|x) + \log p_{\theta_g}(y|x, h^{(i)}) + \log q_\phi(h^{(i)}|x, y) \right) \tag{7}$$

## 2.3 INDEX REBUILDING AND PASSAGE CONCATENATION

**Index Rebuilding.** In previous work, during training, the index of passage embeddings for the KB is often fixed; therefore, the parameters of the passage encoder ($\lambda$) are frozen (Lewis et al., 2020;

Mishra et al., 2022; Lin et al., 2023a). In this work, to study whether JSA-RAG can perform end-to-end optimization of all modules - including the passage encoder, we explore an index rebuilding scheme. During training, we no longer freeze the parameters of the passage encoder and recalculate the passage embeddings in the index using the updated passage encoder at regular intervals. During passage embedding recalculation, the training process waits; the training is resumed after the index update is completed.

**Passage Concatenation.** Note that the prior retriever is improved after end-to-end learning. Inspired by FiD (Izacard & Grave, 2021), we consider a passage concatenation strategy for post-training of the generator while fixing the retriever. The top-$k$ prior retrieved passages are concatenated and append to the context, forming a combined sequence that is fed into the generator for response generation. In this way, the generator is post-trained and in the same way, the generator is used in decoding. This shows the bonus offered by JSA-RAG.

Note that the above two methods are only used in the experiments described in Section 4.5.

## 2.4 DECODING

During testing, we use "Top-$k$ Documents Decoding", following VRAG (Mishra et al., 2022), with $k = 10$. Specifically, given a context $x$, we employ the trained prior retriever to fetch the top-$k$ passages $\{h^{(1)}, \cdots, h^{(k)}\}$. The context $x$ and the retrieved passage $h^{(i)}$ are concatenated and fed into the generator, a beam search is performed to generate the top response $y^{(i)}$, $i = 1, \cdots, k$. We estimate $p(y^{(i)}|x)$ using the product of two terms: $p(y^{(i)}|x) \approx p(h^{(i)}|x)p(y^{(i)}|h^{(i)}, x)$. Finally, we select the response $y^{(i)}$ with the highest estimated probability as the final output for the given context $x$. This is a simplified "Fast Decoding" in (Lewis et al., 2020) and performs well in our experiments.

## 3 EXPERIMENT

To evaluate the effectiveness of the JSA-RAG method, we use VRAG and RAG as baseline end-to-end methods. The evaluation is taken on two tasks - open-domain question answering and knowledge-grounded dialogs. Comprehensive experiments are conducted to evaluate the performance of JSA-RAG, focusing on aspects such as generation quality and retrieval recall. Ablation studies are also conducted to analyze the effects of JSA-RAG in controlling gradient variance and optimizing retrieval efficiency. Specifically, we compare the performance of the posterior retrievers in JSA-RAG and VRAG, as well as fluncations in gradient norms. For all experiments, top-10 retrieval is used for both the prior and posterior retrievers during training and testing.

## 3.1 DATASETS

**Open-domain question answering (ODQA)** requires extensive external knowledge to answer questions, which is the primary task explored with RAG systems. We mainly consider three ODQA datasets: NaturalQuestions (NQ) (Kwiatkowski et al., 2019), TriviaQA (TQA) (Joshi et al., 2017), and MS-MARCO (Bajaj et al., 2016)[1]. For NQ and TQA, we use the Wikipedia December 2018 dump, which contains a total of 24M chunks (passages); for MS-MARCO, instead of using the 10 provided reference passages, we extract its QA pairs and use the MS-MARCO passages from TREC2019 as the KB (Bajaj et al., 2016).

**Knowledge grounded dialogs.** In dialog datasets, we use conversation history turns as $x$ to retrieve relevant passages and take the response of the current turn as $y$, thus constructing $(x, y)$ pairs. We use the OR-QuAC (Qu et al., 2020) and the DoQA (Campos et al., 2020) datasets. OR-QuAC is an open-domain dialog question answering dataset derived from the QUAC (Question Answering in Context) corpus, requiring models to retrieve and reason over external knowledge to answer multi-turn conversational questions. DoQA comprises of open-ended dialog conversations on different domains like cooking, travel and movies. Both datasets follow the knowledge base settings used in VRAG (Mishra et al., 2022). For OR-QuAC, its KB contains 68k passages, while for DoQA, its KB contains 1.2k passages.

---

[1]The MS-MARCO dataset has two versions, and the version we use in this work is MS-MARCO v1.

## 3.2 EXPERIMENT SETTINGS

In open-domain QA tasks, we employ BGE-large (326M parameters) (Zhang et al., 2023) to initialize both the prior and posterior retrievers and use Mistral-7B (Jiang et al., 2023) as the generator, which is fine-tuned with LoRA (Hu et al., 2022; Han et al.) during training. To evaluate end-to-end generation, we use Exact Match for NQ and TQA, and BLEU-1 and Rouge-L for MS-MARCO. For retriever performance, we use Recall@1 and Recall@10 to measure the accuracy of retrieved results for all three datasets.

For knowledge-grounded dialog tasks, we initialize the prior and posterior retrievers with DPR (124M parameters) (Karpukhin et al., 2020) and use GPT-2 (124M parameters) (Radford et al., 2019) as the generator. For end-to-end generation evaluation, we use BLEU-1, BLEU-4 (Papineni et al., 2002), and F1; for retriever performance, we employ metrics the same as in open-domain QA.

Remarkably, evaluating retrievers requires access to the gold passages. For NQ, TQA, and MS-MARCO, there are no gold passage annotations. Therefore, we first perform a posterior retrieval based on BGE-large to retrieve the top 100 passages per question from the KB. Then GPT-4o is used to select the most relevant as the gold annotation. So in testing, the retrieval metrics for NQ, TQA, and MS-MARCO are calculated against a high-quality but machine-generated standard. See Appendix C for details. *It should also be noted that these GPT-4o-generated annotations are never used in model training, but only used in testing to evaluate the prior retrievers (Table 2) and the posterior retrievers (Table 3), for those datasets without gold passage annotations (i.e., NQ, TQA and MS-MARCO in our experiments).*

Technically, we construct an FAISS index for fast retrieval and deploy it as a standalone server. This allows the main program to perform retrieval via API calls. This setup optimizes GPU memory utilization: by eliminating the need to pre-reserve GPU memory for index loading, the main program can allocate more dedicated VRAM to model computations. Notably, the index persists across experiments (when different experiments use the same Wikepedia KB), eliminating redundant embedding recomputation and significantly reducing training time.

Additional training details are provided in Appendix A. The Prompt Template for the generator LLM is shown in Appendix D.

## 3.3 MAIN RESULT

Based on the results in Table 1, JSA-RAG demonstrates significant superiority on dialog datasets (DoQA and OR-QUAC) and open-domain QA tasks, with all evaluated metrics outperforming baselines (vanilla RAG and VRAG).

**End-to-end generation performance.** On DoQA, JSA-RAG achieves a BLEU-4 score of 17.11 (+10.3% relative over VRAG) and an F1 score of 27.84 (+6.9% relative over VRAG), and on OR-QUAC, its F1 score of 18.41 represents a relative 4.4% gain over VRAG, highlighting superior contextual knowledge integration for complex dialog reasoning. In open-domain QA, JSA-RAG excels in TQA with an Exact Match score of 75.23 (+4.1% relative over VRAG) and NQ with an Exact Match score of 51.05 (+4.1% relative over VRAG), indicating robust handling of multi-step questions, and in MS-MARCO with a Rouge-L score of 37.96 (+3.4% relative over VRAG), reflecting fluent and contextually faithful generation. JSA-RAG shows significant improvements on the NQ dataset as well. From these results, we can observe that JSA-RAG completely outperforms the other two methods in end-to-end generation.

**Retrieval performance.** Going beyond end-to-end generation performance, we aim to explore whether the JSA-RAG promotes joint improvement of retriever and generator in end-to-end training. Thus, we conduct experiments to evaluate the retrieval performance. On Table 2, JSA-RAG demonstrates consistent improvements in prior retriever performance across dialogs (OR-QUAC, DoQA) and open-domain QA (NQ, TQA, MS-MARCO) datasets. This reveals that the retrievers can get enhanced in JSA-RAG end-to-end training. On OR-QUAC, JSA-RAG achieves the highest R@1 (39.56, +1.7% relative over VRAG), R@10 (84.76, +1.5% relative over VRAG) and MRR@10 (57.51, +2.7% relative over VRAG), reflecting successful optimizations for dialog retrieval. In open-domain QA, JSA-RAG outperforms baselines on NQ (R@1: 29.23 +8.5%, R@10: 67.27 +5.0%, MRR@10:41.04 +7.4% relative over VRAG), TQA (R@1: 37.39, +1.5% relative over VRAG) and MS-MARCO (R@1: 24.75 +4.5%, R@10:71.32 +3.6% relative over VRAG), indicating stronger

Table 1: Performance comparison of different methods on knowledge-grounded dialog and open-domain question answering datasets.

| Method | Knowledge-Grounded Dialog | | | | | | Open-Domain Question Answering | | | |
| | DoQA | | | OR-QUAC | | | NQ | TQA | MS-MARCO | |
| | BLEU-4 | BLEU-1 | F1 | BLEU-4 | BLEU-1 | F1 | EM | EM | BLEU-1 | Rouge-L |
|---|---|---|---|---|---|---|---|---|---|---|
| RAG | 15.39 | 21.69 | 25.91 | 6.57 | 13.51 | 17.28 | 50.52 | 72.82 | 34.23 | 36.54 |
| VRAG | 15.51 | 21.55 | 26.02 | 6.71 | 13.87 | 17.63 | 49.03 | 72.26 | 34.14 | 36.70 |
| JSA-RAG | **17.11** | **23.36** | **27.84** | **7.76** | **14.59** | **18.41** | **51.05** | **75.23** | **35.28** | **37.96** |

Table 2: Performance evaluation of the prior retrievers for different methods. The base retrievers for knowledge-grounded dialog datasets and open-domain question answering datasets are BGE-large and DPR, respectively.

| Method | Knowledge-Grounded Dialog | | | | | | Open-Domain Question Answering | | | | | | | | |
| | OR-QUAC | | | DoQA | | | NQ | | | TQA | | | MS-MARCO | | |
| | R@1 | R@10 | MRR@10 | R@1 | R@10 | MRR@10 | R@1 | R@10 | MRR@10 | R@1 | R@10 | MRR@10 | R@1 | R@10 | MRR@10 |
|---|---|---|---|---|---|---|---|---|---|---|---|---|---|---|---|
| Base | 31.16 | 77.74 | 48.28 | 58.59 | 83.13 | 66.94 | 26.25 | 63.73 | 37.64 | 33.33 | 66.73 | 44.21 | 10.51 | 36.81 | 17.91 |
| RAG | 38.97 | 83.35 | 55.93 | 67.61 | 87.33 | 74.74 | 27.58 | 66.97 | 39.96 | 36.01 | 69.48 | 46.19 | 23.17 | 68.66 | 36.48 |
| VRAG | 38.91 | 83.48 | 55.98 | 68.01 | 87.49 | 74.78 | 26.95 | 64.04 | 38.21 | 36.81 | 70.07 | 46.73 | 23.68 | 68.81 | 37.53 |
| JSA-RAG | **39.56** | **84.76** | **57.51** | **68.09** | **87.57** | **75.06** | **29.23** | **67.27** | **41.04** | **37.39** | **70.67** | **46.90** | **24.75** | **71.32** | **38.65** |

retrieval of relevant passages. The superiority of JSA-RAG in R@1 and R@10 across multiple datasets indicates that passages filtered by MIS are more helpful in guiding the training of retrievers, compared to simply using a posterior retriever to fetch top-10 passages.

**Comparative analysis.** By combining the analysis of end-to-end generation performance and retriever performance, we find that JSA-RAG comprehensively outperforms both RAG and VRAG. Notably, on TQA and MS-MARCO, although VRAG introduces the posterior retriever and improves retriever performance, its generation performance declines compared to RAG. This exhibits asynchrony in end-to-end optimization, where retriever performance improves while generator performance decreases instead. In contrast, JSA-RAG enables more effective joint optimization between retrievers and generators, achieving simultaneous improvements in both retriever accuracy (e.g., higher R@1, R@10, MRR@10) and generative quality (e.g., superior BLEU-4, F1 scores). This demonstrates that the knowledge pieces selected by JSA-RAG's MIS step can not only enhance retriever performance but also align well with the preferences for generating response, rather than merely maximizing the relevance scores of retrieved knowledge pieces.

## 3.4 ABLATIONS

We conduct ablation experiments on the posterior retriever from two aspects to help understand intuitively why JSA-RAG outperforms VRAG. First, we analyze the performance of the posterior retriever. We test the trained posterior retriever on recall@1, recall@10, and MRR@10 metrics using the OR-QUAC dataset, a moderately scaled dataset with gold passage annotations. Second, we monitor the gradient variation of the posterior retriever during training. This allows us to observe how the gradients fluctuate along the training steps to intuitively compare the gradient variance between JSA and VRAG.

**Performance of posterior retriever.** As shown in Table 3, on the QuAC dataset, JSA-RAG's posterior retriever outperforms VRAG across all evaluated metrics: R@1 (46.91 vs. 45.66), R@10 (91.43 vs. 91.26), and MRR@10 (65.42 vs. 64.52), achieving improvements of 2.7%, 0.2%, and 1.4% respectively. These re-

Table 3: Performance evaluation of the posterior retrievers on the OR-QuAC dataset.

| Dataset | Method | R@1 | R@10 | MRR@10 |
|---|---|---|---|---|
| | DPR | 44.32 | 90.72 | 63.88 |
| OR-QuAC | VRAG | 45.66 | 91.26 | 64.52 |
| | JSA-RAG | **46.91** | **91.43** | **65.42** |

sults indicate that JSA-RAG's posterior retriever captures relevant knowledge pieces more accurately. Presumably, this is because the low-variance gradients allow the posterior retriever to converge more efficiently toward the true posterior distribution during training. Similarly, better quality of passages retrieved by MIS also benefit posterior retriever.

**Variance in gradient norm.** As shown in Figure 2, we record the gradient norms of the posterior retrievers for JSA-RAG and VRAG every 50 steps during the first 4,000 training steps. The gradient

Table 4: Performance of index rebuilding on OR-QuAC (index rebuilt every 100 steps during training).

| Method | Rebuild | BLEU-4 | R@1 | R@10 | MRR@10 |
|---|---|---|---|---|---|
| RAG | ✗ | 6.57 | 38.97 | 83.35 | 55.93 |
|  | ✓ | 8.50 | 47.77 | 89.75 | 65.39 |
| VRAG | ✗ | 6.71 | 38.91 | 83.48 | 55.98 |
|  | ✓ | 8.58 | 48.54 | 90.09 | 65.90 |
| JSA-RAG | ✗ | 7.76 | 40.35 | 84.66 | 57.97 |
|  | ✓ | **10.26** | **49.44** | **90.11** | **66.36** |

Table 5: Performance of passage concatenation on ODQA (top 10 passages concatenated with context).

| Method | Concat | NQ | TQA | MS-MARCO | |
|---|---|---|---|---|---|
|  |  | EM | EM | BLEU-1 | Rouge-L |
| RAG | ✗ | 50.52 | 72.82 | 34.23 | 36.54 |
|  | ✓ | 51.10 | 74.84 | 34.00 | 37.31 |
| VRAG | ✗ | 49.03 | 72.26 | 34.14 | 36.70 |
|  | ✓ | 51.99 | 75.54 | 34.81 | 37.86 |
| JSA-RAG | ✗ | 51.05 | 75.23 | 35.27 | 37.96 |
|  | ✓ | **52.35** | **76.11** | **35.61** | **38.81** |

norms of JSA are of low variance, while the gradient norms of VRAG frequently exhibit "sharp" spikes. This observation confirms that during training, JSA-RAG provides gradients with lower variance, enabling more stable training dynamics and thus yielding higher performance.

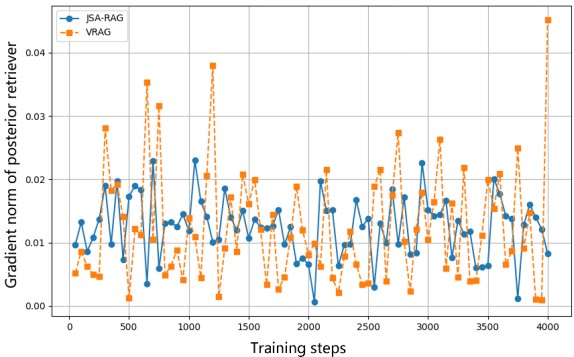

Figure 2: Comparison of the gradient norms from the posterior retriever.

## 3.5 EXPERIMENTS ON INDEX REBUILDING AND PASSAGE CONCATENATION

**Index rebuilding.** In the index rebuilding experiment, unlike the main experiment, we do not freeze the parameters of the passage encoder. Every 100 steps, we recalculate the passage embeddings using the updated passage encoder and rebuild the index. As Table 4 shows, updating the index on OR-QuAC improves all metrics for all methods, with substantial gains in retriever performance, and JSA-RAG remains the top performance. These findings show that JSA-RAG is able to enhance the performance of all system components in an end-to-end training framework. Meanwhile, it should be noted that in our experiments, model training and index rebuilding are serially conducted. In the OR-QuAC (68k passages) experiment, it takes about 3 minutes for 100 training steps, and then index rebuilding takes less than 2 minutes, which hardly affects the training time. However, for larger scale experiments with Wikipedia (24M packages), index rebuilding takes 6 hours with the hardware in our experiments (4*A100 (40G)). When we used a long interval for index rebuiding (we tried 2000-5000 steps in experiments), marginal performance improvements were obtained compared to the results of not using index rebuilding, since the model training behavior does not change much. However, if we reduce the interval for index rebuilding (e.g., every 100 steps), then it will add 200×6 hours with huge time cost (for a total of 20,000 steps for one experiment). Asynchronous algorithm research (parameter training and index rebuilding are performed in parallel) is interesting future work. *It should be noted that such asynchronous algorithm is orthogonal to the core contribution of this work – the developement of JSA-RAG beyond the prior arts which are vanilla RAG (using top-K marginalization) and variational approaches like VRAG.*

**Passage concatenation.** In the passage concatenation strategy, we freeze all parameters except the generator. As shown in Table 5. On the dataset NQ,TQA and MS-MARCO, post-training with the passage concatenation strategy is found to improve performance across all methods. After all methods are subjected to post-training, JSA-RAG still significantly outperforms RAG and VRAG. This robustness shows the practical applicability of the JSA-RAG approach.

## 4    RELATED WORK

**Retrieval-Augmented Generation (RAG).** RAG (Lewis et al., 2020) has become a widely recognized paradigm for combining parametric memory with nonparametric memory. A major challenge in end-to-end optimization of RAG models is that the optimization needs to marginalize over relevant passages, which are modeled as discrete latent variables with no annotations. Atlas (Izacard et al., 2023) studies some ad-hoc loss functions (including the vanilla RAG loss via TKM) to train the retriever jointly with generator, and does not observe significant systematic differences between the different training objectives. This highlights the need for more principled end-to-end training method, which our JSA-RAG addresses. In addition to investigating new training methods for RAG, there are other research activities around RAG. FiD (Izacard & Grave, 2021) presents a new strategy to aggregate and combine multiple passages in decoding. In (Siriwardhana et al., 2023), end-to-end training of RAG is applied to specialized domains such as healthcare and news. CoV-RAG (He et al., 2024) integrates a verification module into RAG and uses verification data to finetune RAG generators, while keeping retrievers frozen. RAT (Wang et al., 2024) combines RAG with chain of thought (CoT) prompting but does not involve any model training. RA-DIT (Lin et al., 2023b) finetunes retrievers and generators separately on a set of multi-task instruction-tuning datasets, which are not end-to-end optimized. *These recent works are orthogonal to and can benefit from JSA-RAG, which focuses on improving end-to-end training of RAG models.*

**Learning with discrete latent-variable models.** End-to-end training of RAG in essence amounts to learning a discrete latent-variable model. A class of maximum likelihood methods consists of the expectation-maximization (EM) algorithm (Dempster et al., 1977) and its extensions. Variational learning optimizes the Evidence Lower Bound (ELBO) instead of directly maximizing the marginal log-likelihood. VRAG and Hindsight, both based on variational learning, use the TKM approximation to optimize ELBO. RetGen (Zhang et al., 2022) uses the REINFORCE trick (Paisley et al., 2012). Stochastic RAG (Zamani & Bendersky, 2024) uses the Straight-Through trick (Bengio et al., 2013). These parameter estimators are known to be biased or have high-variance (Ou & Song, 2020). The JSA algorithm (Xu & Ou, 2016; Ou & Song, 2020) is a stochastic extension of the EM algorithm with impressive performance, where both the E-step and the M-step (as they cannot be performed exactly) are extended by the stochastic approximation methodology, hence called joint SA. *To the best of our knowledge, our work is the first to apply and implement the general JSA to successfully realize more powerful and principled end-to-end training method for RAG.*

## 5    CONCLUSION AND FUTURE WORK

A major challenge in end-to-end optimization of the RAG model is that the optimization needs to marginalize over relevant passages from a knowledge base, which are modeled as discrete latent variables with no annotations. Traditional top-K marginalization and variational RAG (VRAG) suffer from biased or high-variance gradient estimates. In this paper, we propose and develop joint stochastic approximation (JSA) based end-to-end training of RAG, which is referred to as JSA-RAG. The JSA algorithm is a stochastic extension of the EM algorithm and is particularly powerful in estimating discrete latent variable models. JSA-RAG achieves substantial improvements across multiple tasks and datasets, compared to vanilla RAG and VRAG. Notably, it can be seen from Appendix B that the training time cost of JSA-RAG is comparable to RAG and VRAG. Beyond performance evaluation, we demonstrate that JSA-RAG exhibits lower gradient variance than VRAG. We also conduct extensive investigations to further strengthen the JSA-RAG framework, including the index rebuilding and the passage concatenation strategies.

Remarkably, the potential advantage of the JSA-RAG approach in learning discrete latent variable models suggests promising directions for future research, particularly in learning multi-step agents. Basically, RAG can be viewed as a two-step agent (retrieve-then-generate). Currently, the multi-step trajectory of thinking, reasoning, tool use, and planning in building agents needs to be synthesized or annotated. The JSA-RAG methodology investigated in this paper can be extended to learning such multi-step agents. This avenue of exploration is highly promising and warrants further investigation.

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

## A    TRAINING DETAILS

In our experiments, all methods were run for 20,000 steps with batch size being 1 and the best results were recorded. During training, we set different learning rates for the retriever and generator. The loss was optimized using the AdamW optimizer. For the dialog task, using GPT-2 and DPR models, the learning rates for the generator and retriever were set to $1 \times 10^{-5}$ and $1 \times 10^{-5}$, respectively. For the ODQA task with Mistral-7B and BGE models, the learning rates for the generator and retriever were $2 \times 10^{-5}$ and $1 \times 10^{-5}$, respectively. In the passage concatenation experiment, only the generator was post-trained at a learning rate of $1 \times 10^{-5}$ for 10,000 steps. Additionally, the hyperparameter involved in JSA include: MIS sampling steps $m = 50$.

For training with the dialog datasets, we used 8 NVIDIA 3090 GPUs with 24GB VRAM for both training and index storing. For ODQA experiments, training was conducted on 8 A100 GPUs with 40GB VRAM, where 4 GPUs were dedicated to main training and the other 4 to building the index server.

GPT-2 was fine-tuned with full parameters, while the Mistral-7B model was wrapped with a PEFT model for LoRA fine-tuning. The configuration of Low-Rank Adaptation (LoRA) parameters used in this study is presented in Table 6. These settings were applied uniformly across all experiments unless otherwise specified.

Table 6: LoRA Hyperparameter Settings

| Parameter | Value |
|---|---|
| Task Type | Causal Language Modeling (CAUSAL_LM) |
| Rank Reduction Factor ($r$) | 8 |
| LoRA Scaling Factor ($\alpha$) | 16.0 |
| Dropout Probability | 0.0 |
| Bias Training Strategy | None |
| Target Modules | `k_proj, q_proj, v_proj, o_proj, gate_proj, down_proj, up_proj` |

## B    COMPUTATION COST IN TRAINING

To evaluate the computational efficiency of different methods, we measure the training time for 100 iteration steps on the OR-QuAC dataset (batch size = 1). The results are summarized in Table 7. It can be seen that the training time cost of JSA-RAG is comparable to RAG and VRAG.

Table 7: Computation Cost Comparison on OR-QuAC Dataset

| Method | Time (Seconds / 100 steps) |
|---|---|
| JSA-RAG | 198 |
| VRAG | 153 |
| RAG | 148 |

## C    GOLD PASSAGE ANNOTATION FOR RETRIEVER EVALUATION

In testing retrieval performance, some datasets such as NQ, TQA, and MS-MARCO do not have annotations for gold passages in their corresponding KBs. A gold passage refers to a passage containing information capable of answering a question. To obtain such passages or to find them as closely as possible, we first perform a posterior retrieval based on BGE-large to retrieve 100 relevant passages from the knowledge base for each question (the posterior retriever performs retrieval using question-answer pairs to incorporate more information). We then prompt GPT-4o to select the one passage that it deems the most capable of answering the question from the 100 candidates. The specific prompt is shown in Table 8. It should be noted that these GPT-4o-generated annotations are

never used in model training, but only used in testing to evaluate the prior retrievers (Table 2) and the posterior retrievers (Table 3), for those datasets without gold passage annotations (i.e., NQ, TQA and MS-MARCO in our experiments).

Table 8: Prompt for Gold Passage Selection via GPT-4o for retriever evaluation

| Task Type | Prompt Text |
|---|---|
| Gold Passage Selection | Question: {question}, Provided Answers: {answers}. Please select the ID of the passage that best answers the question from the following paragraphs. If there is no passage you think can generate the correct answer, select the ID of the passage that comes closest to answering the question. **Note!!! Only return the value of the passage's id key.** |

# D LLM PROMPT TEMPLATE

During training, to enable the generator to more clearly combine the context and the retrieved passages, we employ different LLM Prompt Templates for different tasks, as shown in Table 9.

Table 9: LLM Prompt Templates Used in Evaluation for Different Tasks

| Tasks | LLM Prompt Template |
|---|---|
| ODQA | [INST] Give a short answer to the Question based on relevant information given in Input. \nInput:{retrieved passage}\nQuestion: {q} \n[/INST]{answer} |
| Dialogs | Input:{retrieved passage}\n <speaker1>{turn1}<speaker2>{turn2} <speaker1>{turn3}<speaker2>{answer} |

