# OpenReview forum: "Improving End-to-End Training of Retrieval-Augmented Generation Models via Joint Stochastic Approximation"
_ICLR.cc/2026/Conference — ICLR 2026 Conference Withdrawn Submission_

### Official Review · Reviewer_iQ64 · 2025-10-29

**Soundness:** 2
**Presentation:** 3
**Contribution:** 2
**Rating:** 0
**Confidence:** 4

**Summary:**

This paper aims to address the challenge of end-to-end optimization in retrieval-augmented generation (RAG) systems. Existing RAG models typically consist of separately trained retrievers and generators. Achieving true end-to-end optimization would require marginalizing over all relevant passages in the knowledge base, which are modeled as discrete latent variables. However, current methods tend to be either biased or exhibit high variance.
To tackle this issue, the authors propose a new training framework called JSA-RAG, which applies the Joint Stochastic Approximation (JSA) algorithm to RAG training. By employing a stochastic EM approach to train a posterior retriever, the model enables genuine end-to-end optimization of RAG.
Experiments conducted on five datasets across two tasks: open-domain question answering (ODQA) and knowledge-grounded dialogue. It demonstrate that JSA-RAG significantly outperforms both Vanilla RAG and VRAG in terms of generation quality and retrieval performance.

**Strengths:**

1. This paper offers a well-founded approach to addressing the gradient estimation challenge arising from discrete latent variables in RAG training. In contrast to RAG (TKM), which suffers from biased gradient estimates, and VRAG, which tends to produce high variance, the proposed JSA-RAG applies the Joint Stochastic Approximation (JSA) algorithm that theoretically ensures unbiased and low-variance gradient estimation for the inference model. Overall, this work presents a well-motivated and insightful research direction.
2. JSA-RAG consistently outperforms RAG and VRAG across all five datasets. It improves both generation quality and retriever performance simultaneously, demonstrating its ability to achieve effective joint optimization.
3. The paper offers a thorough analysis that clearly demonstrates the advantages of the proposed approach. Compared with the frequent sharp spikes observed in the gradient norms of VRAG, JSA-RAG achieves lower variance and exhibits more stable training dynamics. Furthermore, the ablation studies indicate that the posterior retriever trained with JSA surpasses its VRAG counterpart in both recall and MRR performance.

**Weaknesses:**

1. Insufficient Baselines：In the Related Work section, the paper mentions several relevant studies, such as RetGen and Stochastic RAG, and claims that these methods “tend to be biased or have high variance.” However, there is a lack of empirical comparison with these approaches. Including only VRAG as a baseline is insufficient; the experiments should incorporate more baselines for a fair and comprehensive evaluation.
2. Concern about Retrieval Performance Evaluation: Regarding the evaluation of retrieval performance, the paper uses datasets without gold passage annotations and relies on GPT-4o-mini to generate these annotations. However, there already exist datasets with human-annotated gold passages (such as MultiHop-RAG and others). Using such datasets would make the evaluation of retrieval performance more credible and convincing.
3. Mismatch in Model Scale and Analysis Scope：The Dialog datasets have relatively small knowledge bases, whereas ODQA involves much larger ones. However, the ablation studies are conducted only on the OR-QuAC dataset, which differs substantially from real-world RAG scenarios. This raises concerns about whether the experimental results can generalize to larger-scale datasets.
4. Concern about Reported Training Costs: The paper claims that the training cost of JSA-RAG is comparable to that of RAG and VRAG. However, as shown in Table 7, the training cost per 100 steps for JSA-RAG is notably higher than that of the baseline models. Hence, the experimental evidence does not fully substantiate the claim of comparable training costs.

**Questions:**

Please see the weakness

---

### Official Review · Reviewer_98hH · 2025-11-01

**Soundness:** 4
**Presentation:** 3
**Contribution:** 2
**Rating:** 4
**Confidence:** 3

**Summary:**

The paper treats the retrieved passage as a discrete latent variable and seeks to maximize the marginal log-likelihood. Instead of top‑K marginalization or ELBO surrogates (vanilla RAG/VRAG), it applies Joint Stochastic Approximation (JSA) with Metropolis‑Independence Sampling (MIS), using a posterior retriever as the proposal. Accepted samples are used as pseudo‑labels to jointly update the prior retriever, generator, and posterior. To make it scale, prior/posterior probabilities are computed on the union of their top‑k sets from a FAISS index. Experiments on ODQA (NQ, TQA, MS‑MARCO) and dialog (OR‑QuAC, DoQA) show consistent but modest absolute gains in generation and retrieval, plus lower gradient‑variance for the posterior. The paper also analyzes index rebuilding and passage concatenation.

**Strengths:**

- Principled estimator for latent retrieval with lower gradient variance on the posterior retriever; clean algorithmic presentation.
- Consistent gains across five datasets (QA and dialog) with multiple metrics.
- Engineering details (FAISS + union top‑k, index rebuilding, passage concatenation) are useful to practitioners.

**Weaknesses:**

- The paper argues for statistical neatness (low‑variance updates), but does not answer why end‑to‑end retriever–generator training is preferable today versus strong non‑E2E alternatives (e.g., well‑tuned retrievers with instruction‑tuned generators, verification‑augmented pipelines, or separate retriever/generator training).
- Inference uses the same top‑k documents decoding; latency/QPS/VRAM numbers versus baselines are absent. Without speed or cost benefits, the case for E2E training hinges entirely on accuracy.
- Improvements are generally +1–3 points (task‑dependent). For many applications, that uplift may not justify the added training complexity and cost.
- Reported wall‑clock shows JSA slower than VRAG (same order but noticeably higher). The paper lacks scaling curves vs MIS steps m and union top‑k that would help calibrate the practicality.
- ODQA retrieval metrics rely on GPT‑4o‑selected “gold” passages from top‑100; robustness to this proxy is untested (e.g., human‑checked subsets or multi‑gold analyses).
- MIS acceptance/mixing statistics and their evolution are not presented; this is important to understand stability, variance, and compute trade‑offs.
- Limited discussion of modern non‑E2E alternatives (e.g., RA‑DIT‑style decoupled training, verifier‑augmented RAG) under matched compute; such comparisons could change the cost‑benefit picture.

**Questions:**

- Scaling curves: How do accuracy and wall‑clock vary with MIS steps m and union top‑k? Where do returns diminish?
- Sampling behavior: What are MIS acceptance rates over training, and do they correlate with final performance or stability?

---

### Official Review · Reviewer_MLy4 · 2025-11-04

**Soundness:** 2
**Presentation:** 3
**Contribution:** 2
**Rating:** 4
**Confidence:** 3

**Summary:**

The paper addresses the challenge of end-to-end RAG optimization, where marginalizing over latent passages leads to biased or high-variance gradients in traditional methods. The core contribution is JSA-RAG, a framework applying the Joint Stochastic Approximation (JSA) algorithm to obtain low-variance gradient estimates. This is achieved using an auxiliary posterior retriever and MCMC sampling. Experiments show significant performance gains over baselines on multiple benchmarks.

**Strengths:**

1. The paper correctly identifies the core gradient estimation problem and applies a theoretically sound solution (JSA) from the statistical machine learning literature.
2. The method demonstrates consistent and significant performance improvements over strong baselines across a diverse set of tasks and datasets.
3. The work is well-supported by thorough analysis, including gradient norm comparisons and ablation studies on index rebuilding and passage concatenation, which enhance the method's credibility.

**Weaknesses:**

1. The framework introduces an auxiliary inference model and a complex MCMC sampling procedure, which increases implementation and tuning difficulty compared to simpler RAG variants.
2. The method's effectiveness relies on a high-quality posterior retriever to guide the MIS sampler. However, the paper does not analyze the framework's sensitivity to a sub-optimal posterior retriever, which could impact training efficiency and convergence.
3. The iterative MCMC sampling introduces a non-trivial computational burden per training step, raising concerns about its scalability as model and data scales increase.

**Questions:**

1. What was the rationale for selecting the number of MIS steps?
2. The analysis of the results should be deepened. Please connect the theoretical benefits of JSA more directly to why the method succeeds on the tested tasks.

---

### Note · Authors · 2025-11-21

I have read and agree with the venue's withdrawal policy on behalf of myself and my co-authors.